# Canopy Phenology and Meteorology Shape the Seasonal Dynamics in Hydrological Fluxes of Dissolved Organic Carbon in an Evergreen Broadleaved Subtropical Forest in Central Japan

**Siyu Chen [1,\*], Ruoming Cao [2], Shinpei Yoshitake [3] and Toshiyuki Ohtsuka [4]**

1   School of Environment and Life Science, Nanning Normal University, Nanning 530001, China
2   School of Ecology and Environment, Northwestern Polytechnical University, Xi'an 710129, China
3   Faculty of Education and Integrated Arts and Sciences, Waseda University, 2-2 Wakamatsucho, Shinjuku, Tokyo 162-8480, Japan
4   River Basin Research Center (RBRC), Gifu University, 1-1 Yanagido, Gifu 501-1193, Japan
\*   Correspondence: nnnu20190814@126.com

**Abstract:** Seasonal variabilities in hydrological fluxes of dissolved organic carbon (DOC) and their driving factors in the evergreen broad-leaved forest are inadequately understood. To aid this understanding, we conducted a three-year study to examine seasonal changes in DOC concentration and flux in throughfall, stemflow, and litter leachate in an evergreen broad-leaved subtropical forest in central Japan. We specifically addressed (1) how DOC in different hydrological fluxes vary on a monthly to seasonal basis, and (2) how canopy phenology and meteorology shape the DOC concentration and flux of throughfall, stemflow, and litter leachate trends in this evergreen forest. Clear seasonal changes were found in throughfall and stemflow DOC concentration but not in litter leachate DOC concentration; the highest throughfall DOC concentrations were observed in spring (10.03 mg L$^{-1}$ in 2017 and 9.59 mg L$^{-1}$ in 2018, respectively) and the highest stemflow DOC concentrations were observed in summer (13.95 mg L$^{-1}$ in 2017 and 16.50 mg L$^{-1}$ in 2018, respectively). Correlation analysis revealed the monthly throughfall DOC concentration to be positively related to the dry weight of fallen leaves (r = 0.72, $p < 0.05$) and flowers (r = 0.91, $p < 0.05$). In addition, Random Forest models predicted that the dry weight of flowers was a primary driver of throughfall DOC concentration and that the DOC concentrations of stemflow and litter leachate were constrained by the throughfall DOC concentration. DOC fluxes in different hydrological flux were significantly positive related to bulk precipitation amounts and temperature. Moreover, the throughfall DOC concentration had a considerable effect on throughfall and litter leachate DOC fluxes. Over 75% of annual net tree-DOC (throughfall + stemflow) fluxes and more than 70% of the annual litter leachate DOC fluxes were produced in the flowering season. Thus, we speculated that the seasonal phenological canopy changes (leaf emergence, fallen leaves, flowering, and pollen) and the sufficient rainfall had great impacts on the amount and quality of DOC concentrations in the evergreen forest; and, furthermore, that the DOC from different forest hydrological fluxes was a significant fraction of the carbon that accumulates in soils.

**Keywords:** throughfall; stemflow; litter leachate; DOC; canopy phenology

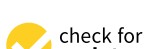



## 1. Introduction

Dissolved organic matter (DOM) is a major variable that modifies the structures and functions of forest ecosystems. It contributes considerably to C (carbon), N (nitrogen), P (phosphorus), and nutrient budgets, and it influences the stability and distribution of C, as well as the microbial activities that occur in the soil. Dissolved organic carbon (DOC) makes up approximately half of DOM; these are the hydrological fluxes of DOC that contain the

crucial chemical components that influence the C biogeochemical cycles in forest ecosystems [1]. In forests, tree canopies are where the first interactions between precipitation and terrestrial ecosystems occur. Precipitation brings nutrient input and is partitioned into throughfall and stemflow according to whether there is a tree canopy. Throughfall is the largest fraction of precipitation (10%–70% of precipitation in temperate and boreal sites, and ~80% of precipitation in the tropics) that falls to the ground between the diverse plant types; in addition, it may or may not make contact with tree canopies [2–5]. Stemflow (generally < 3% of precipitation) is precipitation that moves to the ground via tree trunks or stems [6]. Thus, throughfall and stemflow are the two hydrological processes that transfer the precipitation and DOC from canopies to soil and to the downstream aquatic ecosystems. They are also considered to have demonstrable effects on the hydrology and biogeochemistry of forest ecosystems [4,7,8]. Evergreen broad-leaved forests represent the most dominant warm-temperate forest type at mid-latitudes in the Northern Hemisphere [9]. Thus, the interaction between precipitation and evergreen forest canopies exerts considerable influence over the C cycle of forest ecosystems and the associated downstream processes in the warm-temperate zone.

Forest canopies generate patterns of DOC-enriched hydrological fluxes (both throughfall and stemflow), which are influenced by the interactions between biotic (e.g., canopy phenology) and abiotic (e.g., meteorological conditions) factors [10]. The canopy state (e.g., the leaf area index and canopy structure) and precipitation event type have been found to be crucial indicators of both throughfall and stemflow production [11–13]; for example, the fact that the stemflow amount is larger in a leafless season for various tree species [14–18]. Beyond that, numerous studies have investigated and defined the DOC in throughfall (5–200 mg C $L^{-1}$) and stemflow (1–100 mg C $L^{-1}$) in forest ecosystems at local, regional, and national scales [10,19–26]. Throughfall and stemflow DOC likely percolate slowly through the litter layer and soil matrix where much of it may be consumed or absorbed under low-to-moderate rainfall [23]; however, they can be delivered to the downstream and throughfall precipitation which contributes significantly (30%-80%) to streams under heavy rainfall conditions [27–30]. These findings indicate that tree-DOC (the DOC in throughfall and stemflow precipitation) is a quantitatively significant C flux, which is comparable in magnitude to DOC fluxes in aquatic systems. However, it is poorly integrated into models and C budgets for forest and aquatic ecosystems [10]. Soil is the largest pool of terrestrial organic C in the biosphere, and about 40% of global soil C stocks reside in forests [2,3]. Soil C dynamics could be affected by the selective preservation of labile C inputs from litterfall and roots. The upper organic horizons of podzols are a key source of soil DOC [31]. Michalzik et al. (2001) [24] showed an extensive range in the values of litter leachate DOC fluxes (0.1–0.4 t $ha^{-1}$ $yr^{-1}$) in a studied temperate forest ecosystem; meanwhile, the mean organic carbon pool was up to 156.6 t $ha^{-1}$ in the temperate forest [32]. Although the proportion of litter leachate DOC flux in the soil carbon pool appears small, DOC as a labile C has a higher turnover rate and is considered an early indicator of soil organic carbon (SOC) changes [33–35]; furthermore, it represents 66%–91% of the yearly organic C input into mineral soil [36]. Hence, forest litterfall plays a pivotal role in the interaction between plants and soil, and the DOC from the biological degradation of plant residues and litterfall leaching are important for soil active organic carbon [7,37].

Therefore, identifying the factors regulating the throughfall, stemflow, and litter leachate DOC concentrations and fluxes are essential for evaluating the C cycle in forest ecosystems. DOC concentration and flux in different forest hydrological fluxes are affected by both biotic (i.e., floristic composition, canopy structure, bark morphology, litterfall, and epiphyte) and abiotic (rainfall chemistry, precipitation intensity and frequency, temperature, and dry deposition) factors [10]. Previous studies have indicated that seasonal variability in DOC concentrations can be significant. Van der et al. (2012) [19] indicated that solute leaching from leaves was at its greatest during the leaf senescence season, but at its least during the emergence season. Comiskey (1978) [38] reported that throughfall DOC concentrations in the summer leafed canopy season were over 20 times higher than those in

the winter leafless season. Furthermore, throughfall DOC concentration was significantly related to the average seasonal air temperature in a German deciduous forest [39]. Meanwhile, Levia et al. (2012) [40] found a 50%–60% reduction in stemflow DOC concentration under leafless canopy conditions when compared to fully leafed conditions. In addition, throughfall and stemflow DOC concentrations were reported to be negatively related to precipitation amount and intensity [40,41]. The leaching of organic substances from fresh litter is thought to be the most critical process causing the release of DOC [23,42,43], and the leaf fall that has occurred the entire year in the secondary evergreen forest is different from that in the deciduous forest. The unique leaf fall pattern may cause further changes in the DOC of litter leachate when compared to the deciduous forest.

Although high precipitation, abundant substrate supply, unique canopy phenology patterns, and rapid decomposition characterize the subtropical evergreen forest, the dynamics of DOC in different hydrological fluxes have been less reported, especially in the context of evergreen broad-leaved forest ecosystems [44,45]. Additionally, the majority of previous studies have investigated seasonal alterations to throughfall DOC by focusing solely on the differences between "leafed" and "leafless" periods in temperate forests [46–49]. Furthermore, there is a paucity of studies considering the influence of canopy phenology for throughfall and stemflow DOC in evergreen broad-leaved forests. We have speculated that DOC concentrations and fluxes in different hydrological fluxes exhibit obvious seasonal changes in evergreen forests, even though they did not have clear leafed and leafless seasons. The DOC dynamics in throughfall, stemflow, and litter leachate are governed by different ecological processes that are related to canopy phenology and meteorology. To test these hypotheses, we conducted a three-year study to investigate the seasonal patterns of the DOC concentration and flux of different hydrological fluxes in an evergreen broad-leaved subtropical forest in central Japan. Our objectives were (1) to describe how the DOC in throughfall, stemflow, and litter leachate vary on a monthly to seasonal basis over three years, and (2) to use Random Forest models to predict how canopy phenology and other abiotic factors regulate throughfall, stemflow, and litter leachate DOC characteristics in this evergreen forest over time. These results will facilitate a better understanding of DOC mobilization from the forest canopy to forest floor, which is important to describe the DOC trends of different hydrological fluxes and will help with refining forest C budget models in evergreen forests.

## 2. Materials and Methods

### 2.1. Study Site Description

The study forest is an evergreen broad-leaved forest located in Mountain Kinka ($\sim$60 m a.s.l., 35°26′ N, 136°47′ E) in the Gifu prefecture of central Japan, which is described in detail in Chen et al. (2017a) [50]. In brief, the region is subject to a subtropical monsoon climate with an annual average precipitation of 1861 mm and an annual mean air temperature of 16.2 °C, from 1991 to 2020, and mean temperatures in the coldest month (January) and hottest month (August) of 4.6 °C and 28.3 °C, respectively. The topography of the area is hilly, with young soil, and the bedrock is composed of sedimentary rock on a chert layer [51]. A 0.7 ha (70 m × 100 m) study plot was built on the lower slopes of Mt. Kinka in 1989. In the studied evergreen broad-leaved forest, *Castanopsis cuspidata* was the most abundant overstory tree species in the basal area (87.76%), while *Cleyera japonica* was the most dominant understory subtree species, based on stem number, followed by *Eurya japonica* [50].

### 2.2. Sample Collection

Water samples including bulk precipitation, throughfall, stemflow, and litter leachate were collected biweekly from June 2016 to April 2019. A polyethylene (PE) funnel (diameter: 21 cm) connected to a 20-L PE bottle was used to collect bulk precipitate and throughfall samples. Three bulk precipitation collectors were placed in an open area near the study plot, while nine throughfall collectors were evenly placed in a fixed grid within the study

plot. Nine litter leachate collectors were distributed near the throughfall collectors and using zero tension lysimeter. These lysimeters of 0.0161 $m^2$ area, containing a glass wool plug and draining into a 12 L plastic bottle through a flexible tube, were installed directly underneath the litter layer; the average depth of litter layer was about 3 cm. Stemflow was recorded using 15 rain gauges (HOBO RG3) with a sample reservoir tank (Figure A1); these were installed on the dominant species *C. cuspidata* (various diameters at breast height (DBH) classes: 50, 40, 30, and 20 cm, each with three replicates) and three deciduous trees (*Eleutherococcus sciadophylloides*, *Magnolia obovata*, and *Quercus serrata* in a 40-cm DBH). The rain gauge logged the real-time fluctuation of stemflow flux using the HOBO Pendant Event Data Logger. The collectors of bulk precipitation, throughfall, stemflow, and litter leachate were rinsed with sampling water after the collection of samples, once per month.

Nine litter traps (1-$m^2$ area) were used to collect fallen leaves and flowers from January 2017 to December 2019, once per month. Fallen leaves and flowers, mainly from the dominant species *C. cuspidata*, were dried at 70 °C to measure dry matter weight.

### 2.3. Water Analyses

The water samples were analyzed using the methods described by Chen et al. (2017) [52]. In brief, all water samples were filtered through a 0.45-μm MF-Millipore nitrocellulose membrane before chemical analysis; pH and EC (electrical conductivity) were measured within 24 h. Total organic carbon analyzer was used to measure DOC concentrations (TOC-V, Shimadzu, Japan).

### 2.4. Data Processing

Hydrological fluxes of bulk precipitation (BP), throughfall (TF), and litter leachate (LL) were calculated by dividing the volume (L) by the collection area ($m^2$). The hydrological fluxes of stemflow (SF) per unit stand area ($H_s$, mm) were estimated using Equation (1) according to Strigel et al. (1994) [53]:

$$H_s = (V_s/b) \times (B/S) \tag{1}$$

where $V_s$ refers to the SF volume (L) of the sample tree, b refers to the basal trunk area of the sample tree ($m^2$), B refers to the total basal area of all trees in the study plot ($m^2$), and S refers to the study plot area ($m^2$).

The monthly DOC fluxes of different hydrological flux ($F_m$, kg ha$^{-1}$ month$^{-1}$) were calculated using Equation (2):

$$F_m = hC_m/100 \tag{2}$$

where h refers to monthly water flux (mm) and $C_m$ refers to the average monthly DOC concentration (mg L$^{-1}$).

The monthly net DOC fluxes ($F_n$, kg ha$^{-1}$ month$^{-1}$) in TF and SF were calculated using Equation (3):

$$F_n = h(C_m - C_b)/100 \tag{3}$$

where h refers to monthly water flux (mm) of TF or SF, $C_m$ refers to the average monthly DOC concentration (mg L$^{-1}$) of TF or SF, and $C_b$ refers to the average monthly DOC concentration (mg L$^{-1}$) in BP.

The monthly net DOC fluxes in LL were calculated using Equation (4):

$$F_{nLL} = F_{LL} - F_{nSF} - F_{nTF} - F_{nBP} \tag{4}$$

where $F_{nLL}$ refers to the monthly net DOC fluxes (kg ha$^{-1}$ month$^{-1}$) of LL, $F_{LL}$ refers to monthly DOC flux (kg ha$^{-1}$ month$^{-1}$) of LL, $F_{nSF}$ refers to monthly net SF DOC flux (kg ha$^{-1}$ month$^{-1}$), $F_{nTF}$ refers to monthly net TF DOC flux (kg ha$^{-1}$ month$^{-1}$), and $F_{BP}$ refers to monthly BP DOC flux (kg ha$^{-1}$ month$^{-1}$).

The seasonal DOC fluxes (kg ha$^{-1}$ season$^{-1}$) and annual DOC fluxes (kg ha$^{-1}$ year$^{-1}$) in different hydrological fluxes were calculated by the sum of monthly DOC fluxes in each season and one year, respectively.

### 2.5. Statistical Analyses

All statistical tests were performed using IBM SPSS STATISTICA 22.0 software. Descriptive statistics were calculated for water amount, DOC concentration, and DOC flux in different hydrological fluxes. One-way ANOVA was conducted to determine the statistical differences among monthly DOC concentration and seasonal DOC flux. Correlation analysis was carried out using linear regression analysis. Significant effects were identified as $p < 0.05$ or $p < 0.01$. In addition, we used Random Forest implementation in R package (R Core Team, 2022, v.4.1.3), fitted 500 trees to each target variable, and used the random subset of predictors as candidates for each split of the tree. To express variable importance across all modelled target variables, the relative importance of each predictor variable was weighted by Random Forest predictive ability (% Var Explained) for the target variable.

## 3. Results

### 3.1. Precipitation Partitioning and Seasonal Changes in Hydrological Fluxes

The annual precipitation flux at the studied evergreen forest was 1864 mm in 2017 and 2157 mm in 2018. Most of the fraction of precipitation was partitioned into throughfall (72.6% in 2017; 69.0% in 2018), and the lower fraction of precipitation was partitioned into stemflow (3.6% in 2017; 3.4% in 2018). Litter leachate was attained in up to 80% of precipitation.

The bulk precipitation amount varied seasonally (Figure 1a), the lowest amounts of bulk precipitation always occurred in the winter (December–February), while the highest amounts of bulk precipitation were in the autumn (September–November) of 2017 and the spring (March–May) of 2018, respectively (Figure 1a). The seasonal changes in throughfall, stemflow, and litter leachate were similar to those in bulk precipitation (Figure 1). The amounts of forest hydrological fluxes were always lowest in the winter; however, for throughfall, the highest amounts were found in the summer or the autumn (Figure 1b). Additionally, the amounts of litter leachate peaked in the summer of 2017 and the spring of 2018 (Figure 1d).

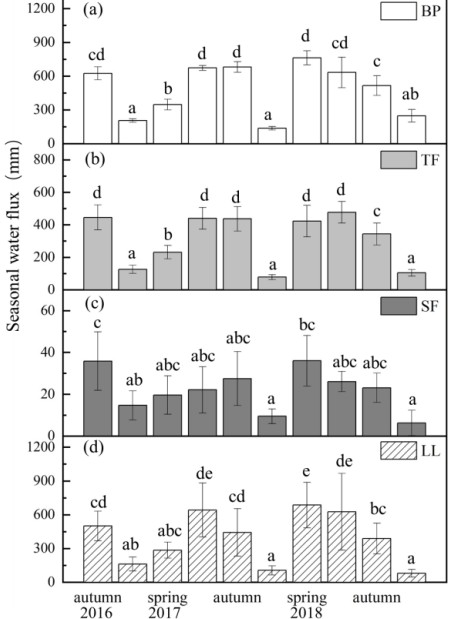

**Figure 1.** Seasonal amounts (mm) of (**a**) bulk precipitation (BP), (**b**) throughfall (TF), (**c**) stemflow (SF), and (**d**) litter leachate (LL), from September 2016 to December 2018. The bars marked with different letters are significantly different at the $p < 0.05$.

### 3.2. Phenological Canopy Changes

The number of fallen leaves and flowers reflected the changes in canopy phenology. In the studied evergreen forest, the canopy phenology was distinct from that of a deciduous forest; despite the absence of a leafless season in the evergreen forest, there was still a period characterized by large numbers of fallen leaves. The largest numbers of fallen leaves and flowers were observed in May (Figure 2a). Figure 2 displays the significant seasonal variations for the fallen leaves and flowers, which both peaked in the spring and were the lowest in the winter (Figure 2b).

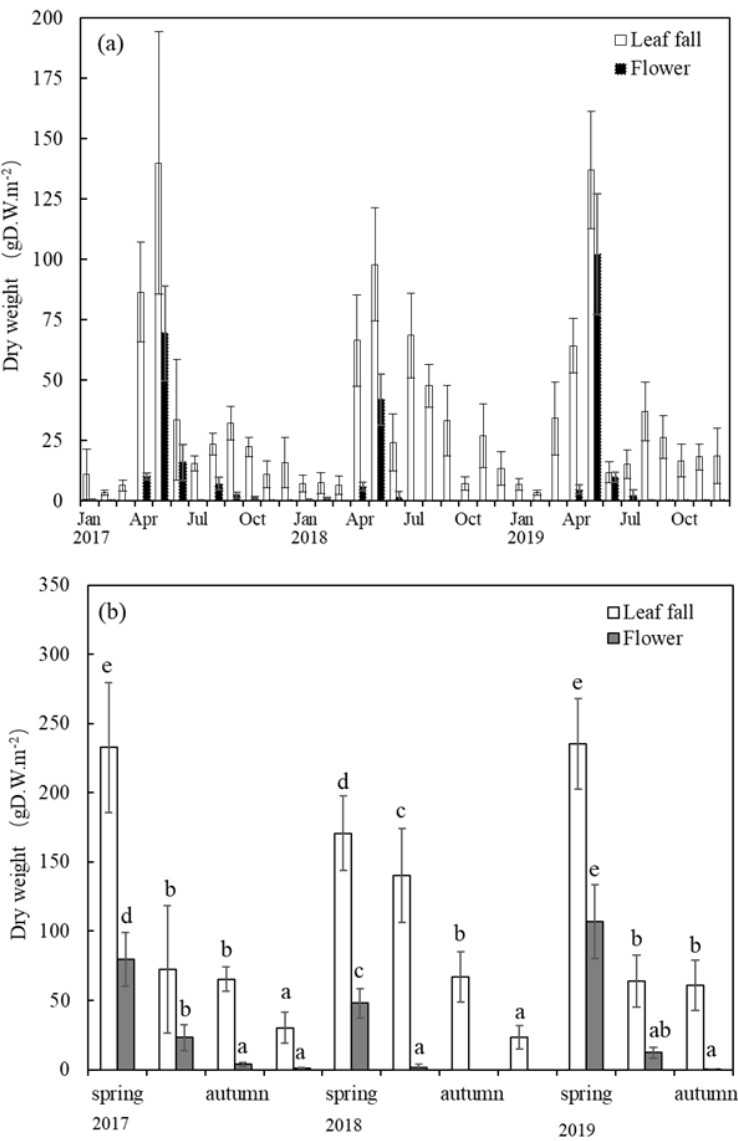

**Figure 2.** Monthly (**a**), and seasonal (**b**), dry weight of leaf fall and flower (gD·W·m$^{-2}$) from January 2017 to December 2019. The bars marked with different letters are significantly different at the $p < 0.05$.

### 3.3. Dynamics of the DOC Concentrations in Different Hydrological Fluxes

No significant difference was found in the yearly concentrations of the bulk precipitation, throughfall, and stemflow. The annual mean DOC concentration in bulk precipitation was 2.8 ± 0.4 mg L$^{-1}$ in 2017 and 2.3 ± 0.2 mg L$^{-1}$ in 2018, respectively. Throughfall, stemflow, and litter leachate were enriched in DOC relative to bulk precipitation. DOC concentrations in throughfall (6.6 ± 1.3 mg L$^{-1}$ in 2017 and 6.1 ± 1.5 mg L$^{-1}$ in 2018) and stemflow (12.0 ± 3.3 mg L$^{-1}$ in 2017 and 12.26 ± 3.7 mg L$^{-1}$ in 2018) were more than

twice and four times as much as those in bulk precipitation, respectively. The annual litter leachate DOC concentration was $19.9 \pm 4.2$ mg L$^{-1}$ in 2017 and $27.0 \pm 8.1$ mg L$^{-1}$ in 2018.

The DOC concentration in bulk precipitation did not distinctly vary by a monthly or seasonal scale; however, it did exhibit significant monthly and seasonal variations in throughfall and stemflow (Figure 3). The highest monthly DOC concentration in throughfall (22.24 mg L$^{-1}$ in 2017 and 20.86 mg L$^{-1}$ in 2018, Figure 3b) and litter leachate ($39.97 \pm 5.85$ mg L$^{-1}$ in 2017 and $43.52 \pm 11.72$ mg L$^{-1}$ in 2018, Figure 3d) were both found in May. For stemflow, the highest DOC concentration was found in May 2017 (16.67 mg L$^{-1}$) and August 2018 (26.00 mg L$^{-1}$), respectively (Figure 3c). Furthermore, seasonal fluctuations in DOC concentrations were fairly comparable in throughfall and stemflow. In addition, the greatest throughfall DOC concentrations were observed in the spring (10.03 mg L$^{-1}$ in 2017 and 9.59 mg L$^{-1}$ in 2018, Figure 3f), whereas the highest stemflow DOC concentrations were detected in the summer (13.95 mg L$^{-1}$ in 2017 and 16.50 mg L$^{-1}$ in 2018, Figure 3g). Litter leachate DOC concentration was lower in winter than in the other seasons. However, no significant difference was found in the litter leachate DOC concentrations between the seasons (Figure 3h).

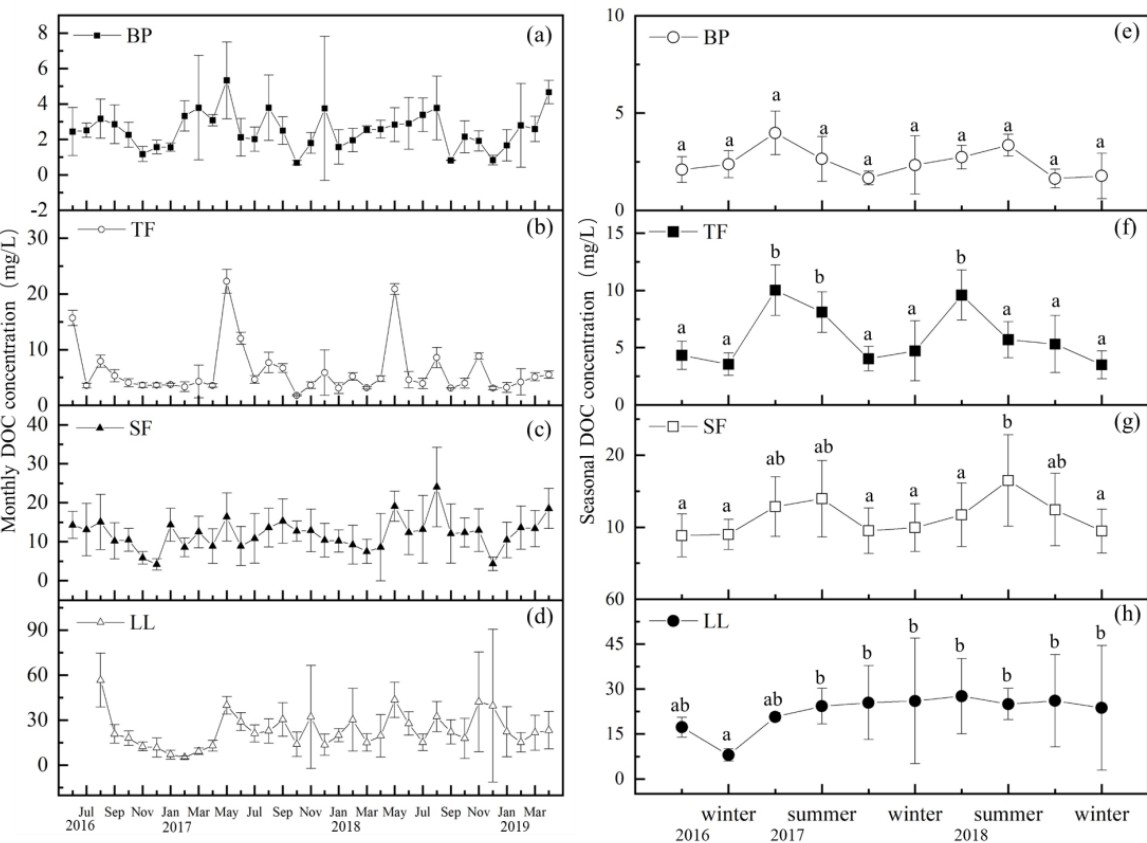

**Figure 3.** Monthly DOC concentration (mg/L) of (**a**) bulk precipitation (BP), (**b**) throughfall (TF), (**c**) stemflow (SF), and (**d**) litter leachate (LL), from June 2016 to April 2019. Seasonal DOC concentration (mg/L) of (**e**) bulk precipitation (BP), (**f**) throughfall (TF), (**g**) stemflow (SF), and (**h**) litter leachate (LL), from autumn in 2016 to winter. The points marked with different letters are significantly different at the $p < 0.05$.

### 3.4. Dynamics of the DOC Fluxes in Different Hydrological Fluxes

The distinct monthly DOC flux changes were detected in different forest hydrological fluxes (Figure 4), while seasonal DOC flux variations were observed in bulk precipitation, throughfall, and litter leachate (Figure 4). In contrast to monthly stemflow DOC fluxes showing erratic change (Figure 4c), monthly throughfall DOC fluxes were higher in either May or June of different years (Figure 4b). Additionally, the highest monthly litter leachate

DOC fluxes were consistently detected in May (Figure 4d). In terms of seasonal DOC fluxes, the lowest DOC fluxes of all forest hydrological fluxes occurred in the winter (December–February; Figure 4), and the highest throughfall and litter leachate DOC fluxes were observed in summer 2017 and spring 2018 (Figure 4).

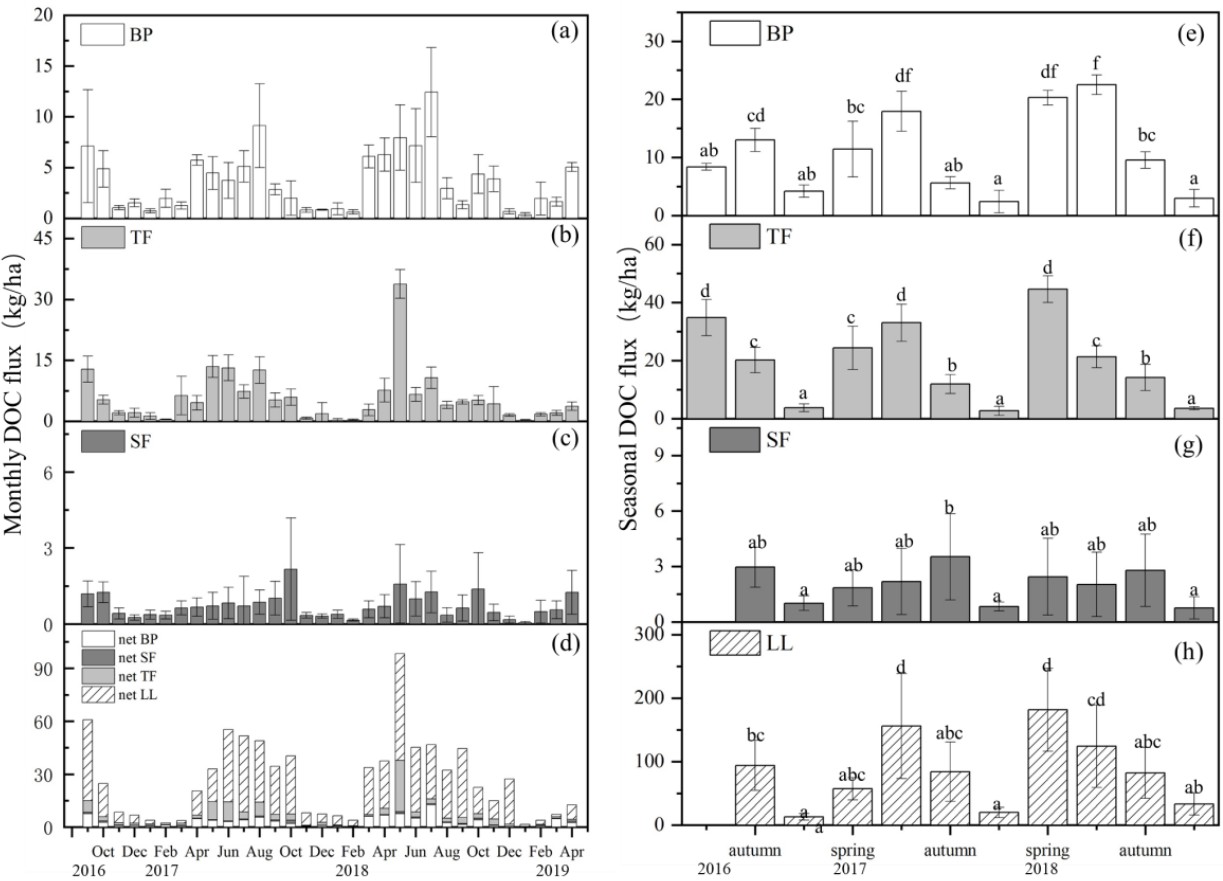

**Figure 4.** Monthly DOC flux (kg/ha) of (**a**) bulk precipitation (BP), (**b**) throughfall (TF), (**c**) stemflow (SF), and (**d**) proportion of net DOC in different water from September 2016 to April 2019. Seasonal DOC flux (kg/ha) of (**e**) bulk precipitation (BP), (**f**) throughfall (TF), (**g**) stemflow (SF), and (**h**) litter leachate (LL), from summer in 2016 to winter in 2018. The bars marked with different letters are significantly different at the $p < 0.05$.

The annual DOC fluxes in different forest hydrological fluxes in 2018 were higher than those in 2017, which is in accordance with the precipitation amounts. The annual DOC fluxes increased in the following order: stemflow, bulk precipitation, and through-fall to litter leachate in both 2017 and 2018 (Figure 5). As shown in Figure 5, the litter leachate DOC fluxes consisted of net DOC fluxes from BP, stemflow, throughfall, and litter leachate. The net DOC fluxes of stemflow, throughfall, and litter leachate were $6.22 \pm 2.61$ kg ha$^{-1}$ yr$^{-1}$, $45.57 \pm 12.47$ kg ha$^{-1}$ yr$^{-1}$, and $227.02 \pm 72.46$ kg ha$^{-1}$ yr$^{-1}$ in 2017, and $5.75 \pm 3.64$ kg ha$^{-1}$ yr$^{-1}$, $51.87 \pm 11.19$ kg ha$^{-1}$ yr$^{-1}$, and $307.24 \pm 79.81$ kg ha$^{-1}$ yr$^{-1}$ in 2018, respectively (Figure 5). There was a slight yearly difference in the DOC proportion from different water fluxes (Figure 5). More than 70% of the annual DOC flux input into the soil were from litter leachate, less than 15% were from throughfall, and around 10% were from BP. Only about 2% of the annual DOC fluxes input into the soil was from stemflow (Figure 5).

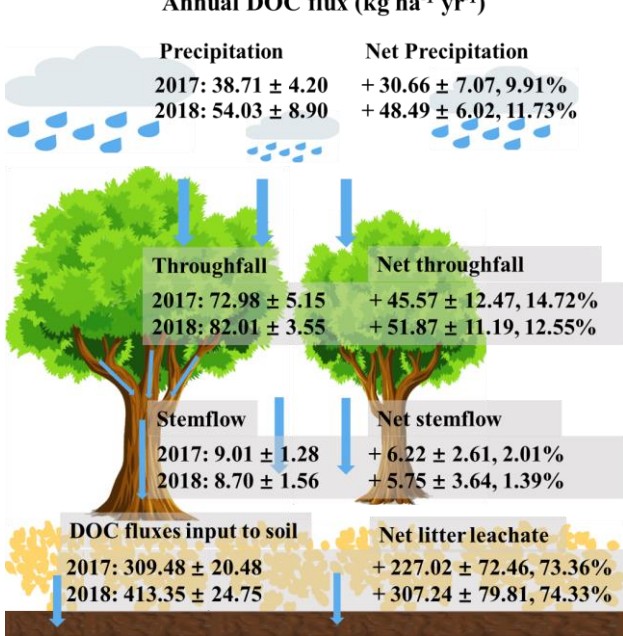

**Figure 5.** Annual DOC flux (kg ha$^{-1}$ yr$^{-1}$), net DOC flux (kg ha$^{-1}$ yr$^{-1}$), and contribution of net precipitation, throughfall, stemflow, and litter leachate DOC fluxes to the annual DOC flux input to soil.

### *3.5. Drivers of DOC Concentrations and Fluxes in Different Forest Hydrological Fluxes*

Correlation analysis revealed that the monthly throughfall DOC concentration was significantly positively related to the dry weight of fallen leaves ($r = 0.73$, $p < 0.01$) and flowers ($r = 0.91$, $p < 0.01$, Figure 6). However, no significant correlations were observed between the stemflow DOC concentration and the dry weight of fallen leaves or flowers (Figure 6). In addition, there was a significantly positive relationship between throughfall DOC concentration and stemflow DOC concentration ($r = 0.52$, $p < 0.05$, Figure 6). Beyond that, the rainfall amount was the main driver of different hydrological DOC fluxes (Figure 6). The monthly throughfall DOC flux was significantly positively correlated to its DOC concentrations ($r = 0.76$, $p < 0.01$, Figure 6).

The Random Forest models were separately calculated according to throughfall, stemflow, and litter leachate DOC concentrations and fluxes; the results indicated that meteorological (rainfall amounts and temperature) and phenological canopy (weights of fallen leaves and flower) variables significantly constrained the throughfall DOC concentrations and fluxes. Among the meteorological and phenological canopy factors, 59.42% of the throughfall DOC concentration variability was explained by the dry weight of the flowers and 24.29% by the dry weight of the fallen leaves (Figure 7a). For the stemflow DOC concentration, the explained variation was associated more with throughfall DOC concentration (26.46%) than with temperature (22.75%, Figure 7b). Similarly, 37.01% of litter leachate DOC concentration variability was explained by throughfall DOC concentration, while 18.28% was explained by the dry weight of the flowers (Figure 7c). Moreover, 37.40% of the throughfall DOC flux variability was explained by throughfall DOC concentration, and 20.69% was explained by the dry weight of the flowers (Figure 7d). Meanwhile, the predominant drivers of stemflow and litter leachate DOC flux were bulk precipitation amount (65.48% for stemflow and 31.74% for litter leachate) and temperature (21.81% for stemflow and 28.12% for litter leachate, Figure 7).

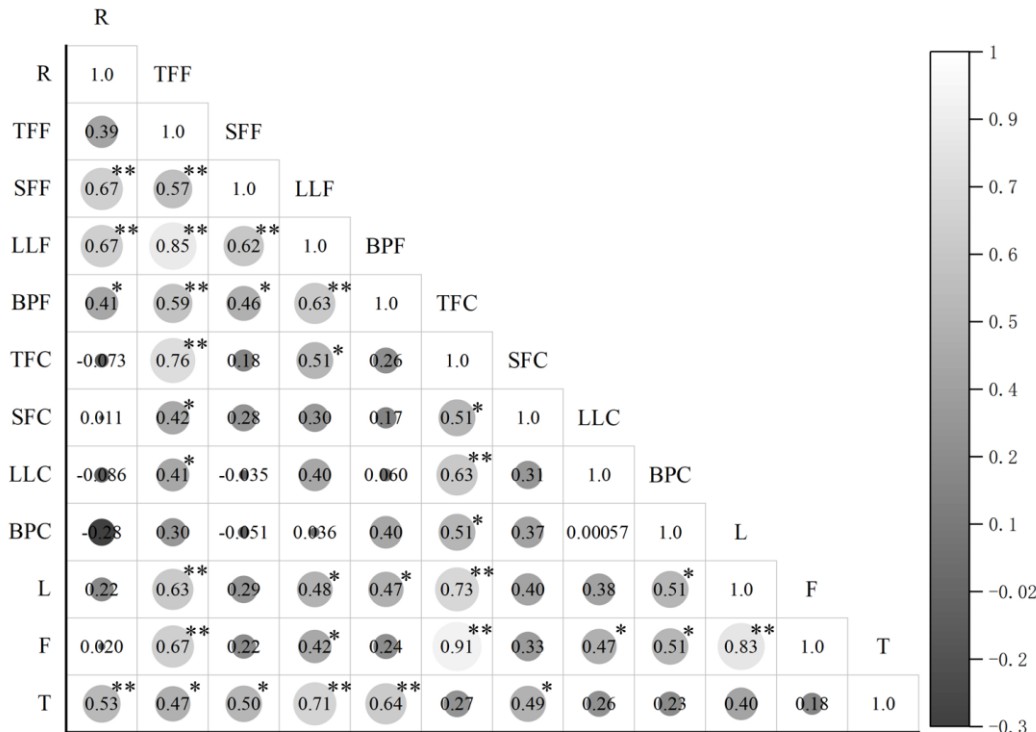

**Figure 6.** Annual DOC flux (kg ha$^{-1}$ yr$^{-1}$), net DOC flux (kg ha$^{-1}$ yr$^{-1}$), and contribution of net precipitation, throughfall, stemflow, and litter leachate DOC fluxes to the annual DOC flux input to soil. The numbers marked with ** and * indicate the correlation was significant at the $p < 0.01$ and $p < 0.05$, respectively.

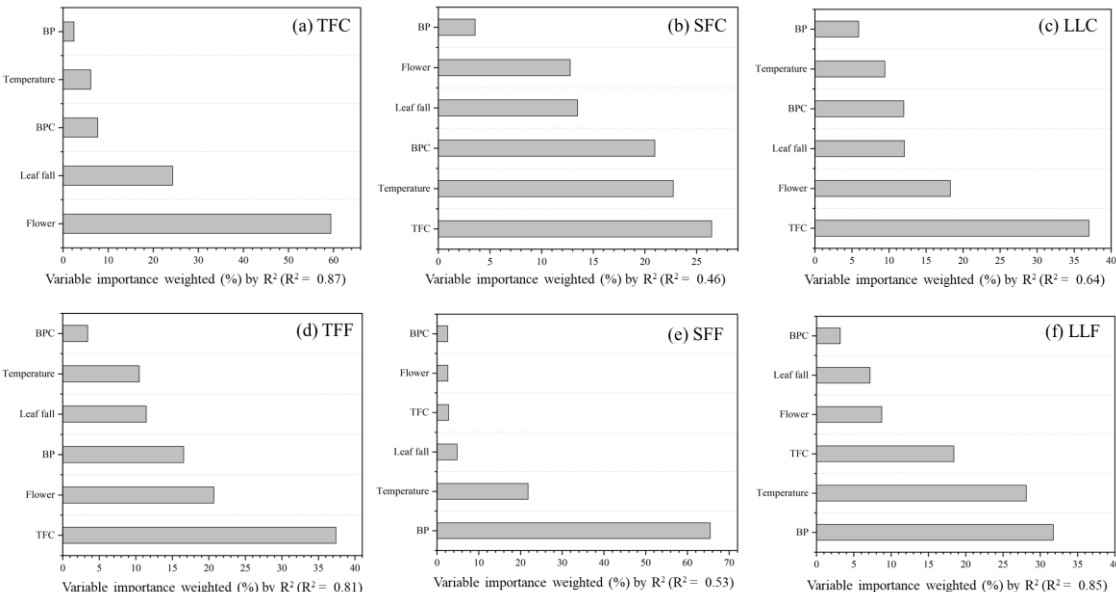

**Figure 7.** Variable importance weighted by R$^2$ for (**a**) throughfall concentration (TFC), (**b**) stemflow concentration (SFC), (**c**) litter leachate concentration (LLC), (**d**) throughfall flux (TFF), (**e**) stemflow flux (SFF), and (**f**) litter leachate flux (LLF). (BPC: bulk precipitation concentration, BP: bulk precipitation amount).

## 4. Discussion

### 4.1. Characteristics of the DOC Concentration in Different Forest Hydrological Fluxes

The annual mean DOC concentrations of the different hydrological fluxes observed at our site were within the range of those reported from other forests [10,23,33,43]. The sources

of DOC were dry deposition wash off, canopy leaching, and incident precipitation; thus, the sources of DOC on canopy surfaces may fluctuate over time in response to seasonal and anthropogenic drivers. The most obvious indicator of seasonal change is the annual cycle of the plants themselves (i.e., canopy phenology) as it is unlikely that flowering, leaf shedding, fruiting, and pollen/seed production will occur simultaneously for each plant within the ecosystem [10]. According to Levia et al. (2011) and Pypker et al. (2011) [4,49], the DOC concentrations in throughfall and stemflow were mainly from canopy (including branches, trunks, and leaves) washing and leaching. Throughfall DOC concentration showed a clear seasonality, i.e., it was higher in the spring than in the other seasons (Figure 5b). Additionally, the monthly throughfall DOC concentrations were positively correlated with the dry weight of the fallen leaves (Figure 5). A prior study noted that the fallen leaves of the dominant species *Castanopsis* in this evergreen forest was the highest in May and was simultaneous with leaf emergence [54]. These findings reflect those of Parker (1983) [55], who also found that the highest concentrations of most of the nutrients in throughfall occurred in the leaf fall season, which could be explained by the seasonal impacts on metabolic processes (such as changes in biological activities and dry depositions) in deciduous forests. Moreover, there is a large amount of pollen production during the florescence period. Although the impact of pollen on throughfall DOC concentration has not been determined, a few studies have reported that pollen may be a significant potential source of DOM and phosphorus [56,57]. Much less research has examined the phenological canopy alterations in throughfall DOC, particularly within an evergreen forest. However, when considering that the DOC throughfall concentration was significantly positively correlated to the dry weight of flowers, and the fact that our models predicted the latter as a predominant driver of changing throughfall DOC concentration in this study, we suggest that the high throughfall DOC concentration could be originating from pollen. As stemflow is regarded as a concentrated version of throughfall [58], and as throughfall DOC concentration is a main factor affecting stemflow DOC concentration (Figures 6 and 7b), the canopy changes are likely having a considerable effect on the stemflow DOC concentration.

DOC in litter leachate is affected by tree-DOC (i.e., the DOC in stemflow and throughfall), litter decomposition, litter moisture, microbial activity, and soil temperature [34,42,59,60]. Litter leachate DOC concentration was generally higher in the leaf fall of deciduous forests [35]; however, the litter leachate DOC concentration had a different pattern in this evergreen forest, whereby it was higher in May of the spring than in the other seasons (Figure 2b). This DOC concentration pattern was accompanied by the highest dry weight of leaf fall in May (Figure 2c). However, a significant positive relationship was not found between the litter leachate DOC concentration and the dry weight of the leaf fall in this study (Figure 6). This supports the results reported by Park and Matzner (2003) [61], who showed that the DOC in litter leachate was derived not only from the recent litter, but also from older organic matter in the lower forest floor horizons. Qualls (2020) summarized that canopy interception—including throughfall chemistry nutrient deposition, pH alteration, and organic substances—may have effects on litter decomposition, implying a complex interaction between litter and microbial activity that produces DOC. Moreover, the litter leachate DOC concentration was positively correlated with the dry weight of the flowers and was linear with the throughfall DOC concentration in this study (Figure 6). Considering that the canopy state is a critical predictor of throughfall and stemflow amounts and of DOC concentration [24,62], its combination with the leaf fall of *Castanopsis* saw a high peak in May, which occurred simultaneously with the leaf emergence period [63,64]. This result indicates that canopy changes will likely affect litter leachate DOC concentrations.

Overall, we purport that, in the studied evergreen forest, the canopy exchanges (leaf emergence, fallen leaves, florescence, and pollen) had a substantial effect on the DOC concentration in throughfall and litter leachate, as well as indirectly impacting the DOC concentrations in stemflow. However, very few studies have yet investigated the DOC dynamics of different hydrological fluxes in evergreen forests because of its inconspicuous phenoseason; therefore, we advise that evaluating the effects of the phenological changes

in tree canopies on the DOC dynamics of different hydrological fluxes is critical in order to enhance our understanding of the source of DOC in evergreen forests.

### 4.2. Characteristics of the DOC Flux in Different Forest Hydrological Fluxes

There is a special variation in throughfall that are related to canopy complexity and meteorological conditions; thus, researchers should be determining an adequate sampling design in order to better assess throughfall [13]. The highest throughfall amount was found in the summer of 2018, while the highest amounts of other hydrological fluxes were observed in the spring (Figure 1); this may be due to the inadequate collectors of throughfall, and this may have led to missing important drip points in this study.

The DOC flux was determined as the product of DOC concentration and hydrological flux amounts for each measurement period; therefore, the highest season of DOC flux occurred when the conditions supported relatively high values of both terms. For DOC fluxes, this occurred most frequently during the spring or the summer in throughfall and litter leachate (Figure 4). Due to the marked phenological shifts occurring in canopy changes during sufficient rainfall in the spring or the summer, the Random Forest models indicated that throughfall DOC concentration significantly constrained the throughfall DOC flux (Figure 6d). It was reported in previous studies that the DOM in throughfall not only passes the forest floor without changes in chemical composition, but also brings throughfall decomposable C compounds. These processes most likely function as cosubstrates or promoters for the decomposition and mineralization processes of the organic matter on the forest floor, thus leading to increased fluxes in DOC with forest floor leachates [33,65]. This supports the result that DOC fluxes in throughfall positively affect litter leachate DOC fluxes (r = 0.85, *p* < 0.01, Figure 6). Additionally, the bulk precipitation and temperature factors were the dominant drivers of stemflow and litter leachate DOC flux; this result might be due to the production of DOC via temperature-sensitive mechanisms, such as litter decomposition, microbial biomass turnover [61], and sufficient water inputs during the summer. This led to proportional increases in DOC fluxes with increasing hydrological fluxes; and, thus, a water limitation on DOC vertical mobilization. Meanwhile, the DOC fluxes in winter were much lower than those observed in other seasons in this study due to water limitation (Figure 4), which suggests a production limitation during the winter. Furthermore, over 75% of annual net tree-DOC fluxes were from the spring and the summer (Figure 4), and the annual DOC fluxes from litter leachate input to the soil were approximately 70% (69% in 2017 and 71% in 2018) in the spring and the summer (March–August, Figure 4), which are the flowering seasons.

Furthermore, the DOC fluxes from litter leachate into the soils with litter-derived DOC fluxes usually exceeded the tree-DOC fluxes [66]. Litter leachate DOC fluxes increase with rainfall, as is shown in the present and previous studies [34]; therefore, the annual litter leachate DOC fluxes in 2017 were lower than those in 2018. Additionally, the yearly litter leachate DOC fluxes at 309–413 kg ha$^{-1}$ yr$^{-1}$ found in the current study were lower than those reported by Fujii et al. (2009) [43], who obtained their results from a tropical forest (470–562 kg ha$^{-1}$ yr$^{-1}$). This happened due to the higher rainfall amounts in their study site (2187–2427 mm yr$^{-1}$); however, their results were comparable to those from boreal (200–480 kg ha$^{-1}$ yr$^{-1}$) and temperate forests (50–200 kg ha$^{-1}$ yr$^{-1}$) [22,43,59].

### 4.3. The Ecological Implications of DOC Flux in Subtropical Secondary Evergreen Forests

The average annual DOC fluxes from litter leachate input to the soil were 361.4 kg ha$^{-1}$ yr$^{-1}$ in the current study. The major DOC flux contribution was found to be from net litter leachate, which was more than 73%, and approximately five times, that of net tree-DOC fluxes (Figure 5). In mature temperate soils, C accumulation is slow (10–120 kg ha$^{-1}$ yr$^{-1}$). In contrast, recently disturbed temperate soil rates ranged from 50 to 400 kg ha$^{-1}$ yr$^{-1}$ [5]. Comparing these accumulation rates to the total amount of DOC fluxes delivered to the soil (309.48 ± 20.48 kg ha$^{-1}$ yr$^{-1}$ in 2017 and 413.35 ± 24.75 kg ha$^{-1}$ yr$^{-1}$ in 2018), it is apparent that the DOC from different water fluxes could provide a significant



fraction of the C that accumulates in soils. Moreover, this study shows that DOC flux variation is well described by the interactions between precipitation flux and phenological canopy variation, which was higher in the spring or the summer, in warm-temperate evergreen forests. These seasonal trends suggest that the predicted increases in precipitation at mid-to-high latitudes in the Northern Hemisphere [67] can result in a proportional increase in DOC hydrological fluxes in the spring or the summer of warm-temperate evergreen forests, and it may also consequently influence the C budget in the soil and downstream currents. These results imply that the climate change effects to warm-temperate evergreen forest DOC fluxes will depend on the seasonal change in precipitation. Soil C is one of the largest organic C pools on Earth [5]. Thus, understanding how soil C will respond to land use, climate, and other environmental changes is critical to predicting the future C budget and climate of the planet. The role of DOC fluxes in soil C accumulation has been overlooked in both natural and urban settings. Future research is required to address this critical knowledge gap.

## 5. Conclusions

This study identified that the dynamics of DOC concentrations and fluxes in throughfall, stemflow, and litter leachate differed in an evergreen broad-leaved forest in a subtropical zone from those in the intensively investigated deciduous forests in temperate zones. Litter leachates were significantly enriched in DOC when compared to stemflow and throughfall. Throughfall and stemflow DOC concentrations exhibited markedly seasonal variations, and the dry weight of flowers was a primary driver of throughfall DOC concentration; meanwhile, stemflow and litter leachate DOC concentrations were constrained by throughfall DOC concentration. These results demonstrated that the DOC concentration in different water fluxes positively responded to the phenological changes in tree canopies (fallen leaves, flowering, and pollen). Moreover, over 75% of the annual net tree-DOC fluxes were from the flowering season, and more than 70% of the annual litter leachate DOC fluxes were produced in the spring and summer of this subtropical secondary evergreen forest. Consequently, we speculate that the seasonal phenological canopy changes may affect the quantity and quality of DOC in different forest water fluxes, even though there are no clearly demarcated leafed and leafless seasons in the studied evergreen forest. The average annual DOC fluxes in litter leachate that were delivered to the soil were found to be up to 361.4 kg ha$^{-1}$ yr$^{-1}$. More than 70% were from net litter leachate, and approximately 13% were from net throughfall. Around 10% were from BP, and only about 2% were from net stemflow. This implied that the DOC from litter leachate was a significant fraction of the C that accumulates in soils.

**Author Contributions:** Conceptualization, S.C.; methodology, S.Y.; formal analysis, S.C.; investigation, S.C. and R.C.; writing—original draft preparation, S.C.; writing—review and editing, T.O.; visualization, S.C.; supervision, T.O. All authors have read and agreed to the published version of the manuscript.

**Funding:** This research was supported by the National Natural Science Foundation of China, Grant No. 32001172, and the Guangxi Natural Science Foundation under Grant No. 2022GXNSFBA035461.

**Data Availability Statement:** Not applicable.

**Acknowledgments:** We wish to thank the forest administration of Gifu for granting us access to the site. In addition, we are grateful for the support regarding field surveys from the members of the Ohtsuka Lab. Furthermore, we give thanks to Fusheng Li, River Basin Research Center, Gifu University, for providing the TOC analyzer and for suggestions regarding the DOC method.

**Conflicts of Interest:** The authors declare no conflict of interest.

**Appendix A**

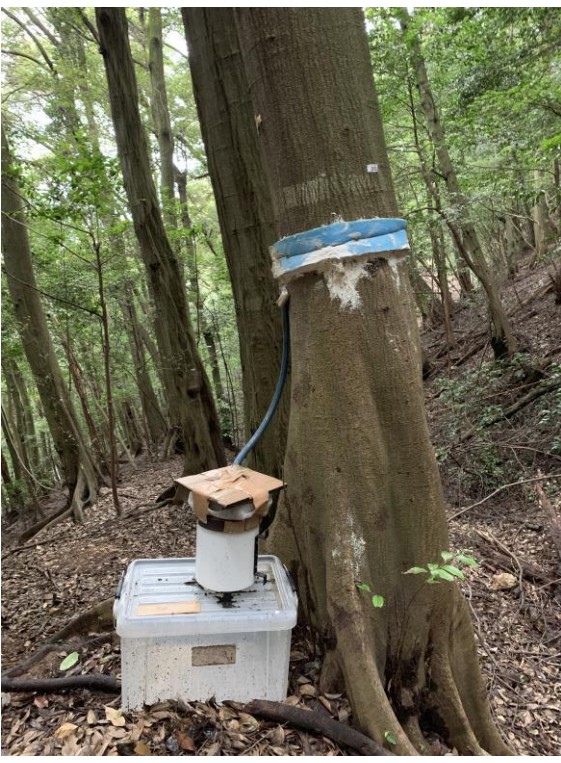

**Figure A1.** Collector of stemflow.

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
