# Peer review of "Canopy Phenology and Meteorology Shape the Seasonal Dynamics in Hydrological Fluxes of Dissolved Organic Carbon in an Evergreen Broadleaved Subtropical Forest in Central Japan"

_forests, doi:10.3390/f14051013_

Round 1

Reviewer 1 Report

This manuscript describes the magnitude and timing of forest hydrological and DOC fluxes over a period of three years to investigate the effects of phenology on nutrient cycling and water quality in an evergreen broad-leaved subtropical forest in central Japan. The study methods and the writing (particularly in the intro and discussion) requires some revision for clarity and grammar. Recommendations provided line-by-line below:

Line 6: Consider rephrasing for clarity. Perhaps something like, “Dissolved organic carbon (DOC) makes up approximately half of DOM. The hydrological fluxes of DOC are crucial chemical components that influence the C biogeochemical cycles in forest ecosystems”

Line 14: At what scale and in what circumstances in throughfall “91% of precipitation”? In some forests, it is much less, in others, this is correct. Please revise.

Line 21: Consider rephrasing for clarity, for example, “The forest canopy generates patterns of DOC-enriched hydrological fluxes (throughfall and stemflow) that are influenced by interactions between biotic (e.g. canopy phenology) and abiotic (e.g. meteorological conditions) factors”

Line 29-31: What is meant by "other biotic factors"? In fact, the reference used here is from over a decade ago. The authors would benefit from updating this statement with information from the most recent reviews:

https://link.springer.com/chapter/10.1007/978-3-030-29702-2_4

https://link.springer.com/chapter/10.1007/978-3-030-29702-2_6

Line 37-38: It would be helpful to provide more specific information about how stemflow affects downstream aquatic ecosystems.

Line 50: Why is there no citation for the stemflow fraction (<3%)? You could cite this mini-review: https://doi.org/10.3389/fpls.2018.00248

Line 81-82: Consider rephrasing for clarity. For example “Although tree-DOC (DOC in throughfall and stemflow) is a quantitatively significant C flux, comparable in magnitude to DOC fluxes in aquatic systems, it is poorly integrated into models and C budgets in terrestrial and aquatic ecosystems.”

Line 105-107: please clarify what is meant by 'conversely related' - This term is confusing.

Line 147: Nine throughfall collectors are very few given the complexity of spatial patterns in this flux. Moreover, few throughfall collectors on a fixed grid may miss important drip or dry points. Please acknowledge and discuss these issues. (e.g., https://link.springer.com/chapter/10.1007/978-3-030-29702-2_6)

Line 152-156: How was the tipping bucket drainage (from stemflow) routed to a collection bin? Perhaps an image of this design would help clarify? Moreover, how often was this system cleaned? Tipping buckets accumulate substantial organic debris which could affect stemflow solute concentrations and interfere with stemflow drainage (possibly increasing stemflow DOC concentrations while decreasing stemflow water reaching the bins). Please provide more information about this.

Lines 211-213: Please provide more information regarding the Random Forest analysis. For example, what were (if there were any) the preprocessing steps taken to prepare the data for the Random Forest analysis, including any feature selection, normalization, or imputation techniques that were used? What performance metrics were used to evaluate the accuracy and generalization ability of the Random Forest model, such as accuracy, precision, recall, F1 score, or area under the curve (AUC)? Currently the authors simply reference the software used.

The results are mostly straightforward, well presented in high quality figures, and, if anything, there may be a touch too much detail (7 figures and a table). I recommend the authors contemplate how they may tell the story of their results in a way that puts less of an interpretive burden on the reader.

Lines 326-337: These lines erroneously reference a figure that does not exist (Figure 10a-d).

The first two discussion sections are a bit superficial, i.e., the majority of the text is just a comparison of their observations to observations from other sites. Such lengthy comparisons could be made in a single statement, like “The water and DOC fluxes observed at our site were within the range of those reported from other forests.” With that reduced to a single statement, I ask the authors to please discuss the novel insights that their study contributes to theory. For example: Do any novel hypotheses arise from this work to direct future work? What novel mechanisms may have been described through these observations? How does this work, which includes interesting phenological information, inform us regarding the source of DOM (canopy v. deposition)?

Author Response

Dear reviwer:

     We would like to express many thanks to your professional review work on this manuscript and sincere suggestions. As you concerned, there are several problems that need to be addressed. According to your valuable suggestions, we have made extensive corrections to our previous manuscript, the detailed corrections are listed in the attachment.

     Please see the attachment。

Sincerely,

Chen

Reviewer 2 Report

Reviewer Response

Manuscript Title: Canopy Phenology and Meteorology Shape the Seasonal Dynamics in Hydrological Fluxes of Dissolved Organic Carbon in an Evergreen Broadleaved Subtropical Forest in Central Japan

Journal: Forests

Code: forests-2292968

General:

This manuscript by Chen et al. addresses the concentrations and fluxes of DOC in an evergreen subtropical forest. The fluxes encompass the pools in precipitation, throughfall, stemflow and tree litter, but does not include the soil pool below the litter layer. Although the manuscript is mostly descriptive, I believe it offers important insight to an understudied area. In general, the methodology and execution of the experiment are solid, however the introduction and discussion fail to convey the importance of the study or present the main results in a way that convey a clear message.

Major weaknesses:

1.    There is no clear line of thought in the Introduction. I suggest the authors minimize the number of specific examples (L53-L91) in the introduction and rather try to convey the main knowledge in the field and why it is important in a broader context.

2.    In the Introduction, please clearly describe the different pools of DOC considered in this study and how they compare to the total pool of C in the ecosystem. In the present manuscript soil OC is not mentioned although it is likely the largest pool of OC and DOC in the ecosystem. While tree litter is the source of DOC to deeper soil layers, most of the C storage is in the mineral soils.

3.  The different fluxes of DOC are interesting, but since they are all strongly influenced by the amount of precipitation, it is unnecessary and misleading to correlate them to potential drivers and other DOC fluxes. I suggest you remove the flux measurements from Figure 9.

4.    Figure 10 is missing from the manuscript and is one of the more important results of the manuscript.

5.  At present state the discussion is a long list of comparisons with other types of ecosystems in different climates. This is relevant for our understanding of the global C pools, but the discussion needs to exemplify why this important at the local scale as well. I believe the discussion needs to be completely rewritten to clearly present the relevance of the paper and how this evergreen forest functions from the C perspective.

Minor weaknesses

6.    Although 3 years of data have been taken, in the majority of analysis is done on only 2 years of data, why is this?

7.  (l150) How and when were the lysimeters installed and how deep is the litter layer? Also, please describe the soil type.

8.     (L133) Please add information on the long-term climate in the area, at the moment means from only 2 years are presented.

9 While the data is well presented, several results are presented in 3-4 places. Presenting both the monthly and seasonal plots as well as describing them in tables and the text is redundant. I suggest the seasonal plots are removed and the values removed from the text in the Results section.

10.     Please add the p and F statistics in the results section where applicable.

11.     L281, please add the year after winter in the figure text.

12.     Figure 6. Why are the colors different in 6b than in the other figures? Also why is net LL presented separately in 6d.

13. You see a clear difference in DOC contributions from leaves and flowers. Please discuss these differences more in the discussion.

Author Response

Dear reviewer:

     We would like to express many thanks to your professional review work on this manuscript and sincere suggestions. As you concerned, there are several problems that need to be addressed. According to your valuable suggestions, we have made extensive corrections to our previous manuscript, the detailed corrections are listed in the attachment.

Sincerely,

Chen
